# Real-Time Ship Segmentation in Maritime Surveillance Videos Using Automatically Annotated Synthetic Datasets

**DOI:** 10.3390/s22218090

**Published:** 2022-10-22

**Authors:** Miguel Ribeiro, Bruno Damas, Alexandre Bernardino

**Affiliations:** 1ISR—Institute for Systems and Robotics, 1049-001 Lisboa, Portugal; 2CINAV—Centro de Investigação Naval, 2810-001 Almada, Portugal

**Keywords:** computer vision, ship detection and segmentation, real-time processing, synthetic datasets, maritime surveillance

## Abstract

This work proposes a new system capable of real-time ship instance segmentation during maritime surveillance missions by unmanned aerial vehicles using an onboard standard RGB camera. The implementation requires two stages: an instance segmentation network able to produce fast and reliable preliminary segmentation results and a post-processing 3D fully connected Conditional Random Field, which significantly improves segmentation results by exploring temporal correlations between nearby frames in video sequences. Moreover, due to the absence of maritime datasets consisting of properly labeled video sequences, we create a new dataset comprising synthetic video sequences of maritime surveillance scenarios (MarSyn). The main advantages of this approach are the possibility of generating a vast set of images and videos, being able to represent real-world scenarios without the necessity of deploying the real vehicle, and automatic labels, which eliminate human labeling errors. We train the system with the MarSyn dataset and with aerial footage from publicly available annotated maritime datasets to validate the proposed approach. We present some experimental results and compare them to other approaches, and we also illustrate the temporal stability provided by the second stage in missing frames and wrong segmentation scenarios.

## 1. Introduction

These days, maritime surveillance is a key piece of every country’s policy. The safeness and secure use of the oceans must be guaranteed in a time when every inch of the ocean is used due to the globalization of our world. Surveillance allows the control of all kinds of illegal activity (smuggling, piracy, and others). It is an important tool in search and rescue missions and, moreover, ensures that environmental regulations are applied, preventing excessive fishing and all kinds of activities that have a severe impact on ecosystems.

Seeing the ocean as a highway and vessels like traffic, maritime transport becomes the backbone of international trade and the global economy, being responsible for around 80% of global trade by volume and over 70% of global trade by value [1]. Worldwide ports must handle the constant flow of incoming and outgoing ships.

A wide variety of sensors have been used for maritime surveillance. The traditional approach implies the use of satellites, vessels and aircraft. Satellites are costly and do not have the necessary flexibility for the job. Traditional aircraft are equipped with heavy radars [2] like Synthetic Aperture Radar (SAR) [3] or other technologies such as Automatic Identification System (AIS) and Vessel Monitoring Systems (VMS). AIS and VMS resort to very high frequency (VHF) and global positioning systems (GPS) to wirelessly transmit the identity and the current position of a ship. However, not all ships are compelled to carry transponders, and in cases of illicit activities, they can be intentionally turned off to avoid radar detection. Furthermore, this technologies remain expensive and in need of large infrastructures like heavy vehicles, satellites or coastal bases. In recent years, unmanned aerial vehicles (UAVs) are increasingly being used in maritime surveillance missions, due to their versatility and reduced operational cost. They can carry a variety of sensors on their payload, from LiDAR to hyperspectral sensors, as they continue to become smaller and more compact. Among these, we can find the ubiquitous visible spectrum camera (VSC), a low cost solution that typically presents low weight, power and space requirements when compared to other technologies.

This paper presents a new approach to detect and segment vessels in maritime scenarios, using images taken from a standard visible spectrum onboard camera like the ones shown in Figure 1, that can run in a real-time manner on a commercial off-the-shelf embedded computational unit on board the UAV, thus being capable of extracting reliable information from the captured images and pass it to a human operator, discarding the situations considered irrelevant.

Image segmentation, i.e., classification of the image on a per-pixel basis, where a class is assigned to each image pixel, provides some information that is difficult to obtain from a pure image detection algorithm, which only identifies the presence of certain classes of objects and locates the detected instances in the image. Besides ship size, a segmented image can convey information regarding the ship shape, that can be later be automatically compared and matched to the ship Vessel Maritime Mobile Service Identity, given by the AIS signature provided by the ship transceiver [5]. Additionally, the obtained ship silhouette, provided by the segmentation algorithm, can be used to estimate the ship orientation and route. Examples of maritime images together with corresponding ship segmentation masks are presented in Figure 2.

The vessel segmentation algorithm proposed in this paper allows UAVs to only send visual feedback (together with corresponding location) to the human operator when there is enough confidence in the presence of a suspicious activity, based on the acquired images. This method has two major advantages: on one hand, a single human operator can potentially deal with several UAVs simultaneously deployed; on the other hand, a large surveillance area can be covered, since each UAV only sends information considered relevant and thus a larger distance from the ground control station can be achieved due to smaller bandwidth requirements, as a full video stream transmission is no longer required.

In the last decade the advent of deep neural networks has significantly changed the methods used for image detection and segmentation, and nowadays the use of deep convolutional networks (CNNs) is the standard approach for these tasks. There have been many works on ship detection in airborne images [7]: in [8], for instance, the authors introduce a new method to detect ships in maritime surveillance scenarios that relies on CNNs to perform robust detection even in adverse conditions. The proposed detection system is composed of two main parts: the first is responsible for generating image patches and the second consists of a CNN to classify every image patch generated by the first stage. In [9], the authors present an application of a Faster R-CNN in Synthetic Aperture Radar (SAR) based ship detection that generates bounding boxes and a constant false alarm rate (CFAR) detector that evaluates the bounding boxes with a low detection score, by calculating the probability of false alarm associated with a detection threshold. Image acquisition and processing with SAR sensors, however, requires additional computation resources and has limited data acquisition speed and real-time capabilities compared to RGB images acquired from UAVs. Ref. [10] presents a framework called Rotation Dense Feature Pyramid Networks (R-DFPN) capable of detecting ships in the ocean and in ports with RGB images. The Feature Pyramid Network gathers multilevel information and achieves state-of-the-art results in small object detection tasks, and in R-DFPN, this architecture is improved by adding connections between feature maps.

Since images taken from video sequences during maritime surveillance missions are highly correlated, ref. [11] propose a tracking method based on correlation filters complemented with image segmentation. A blob analysis stage is added to compensate for drifts in the correlation filter, which allows to re-center the target into the tracking window. Since most of the images are usually composed of the target vessel and the ocean, a binarization approach is adopted to segment images. The segmentation stage resorts to Otsu’s method [12] to separate bright and dark parts of the image, which should correspond to ships and the surface of the ocean, respectively. In [13], a robust autonomous detection approach for airborne surveillance in maritime scenarios combines the use of CNNs to generate region proposals with a multiple hypothesis tracker (MHT) to associate them. The network relies on a modified version of DetectNet to detect, track and predict future positions of objects of interest. More recently, ref. [14] analyzes the effectiveness of learning temporal features to improve detection performance in video sequences captured by small aircraft. A new method is introduced to robustly detect objects in airborne maritime surveillance video sequences affected by glare, wakes and wave crests. They resort to a convolutional long short-term memory (convLSTM) to learn relevant visual and temporal features to improve detection in airborne video sequences. The convLSTM is a modified version of a traditional long short-term memory (LSTM), where the input-to-state and state-to-state multiplications are replaced by convolutions. A pre-trained CNN is used to generate features. The convLSTM is placed after the CNN and receives the computed features in a given moment, having at the same time access to the cell and the data from previous moments. The system performance achieved better results in adverse conditions than in scenarios where the ship is totally visible.

Similarly to detection, in recent years CNNs have also become the dominant method for image segmentation, and recent state-of-the-art deep neural networks for image segmentation comprise the Fully Convolutional Network (FCN) for Semantic Segmentation [15], the U-Net [16], a U-shaped encoder–decoder network architecture, SegNet [17], that also has a encoder–decoder architecture, Mask R-CNN [18], a CNN based on Faster R-CNN [19] that can simultaneously detect objects and perform semantic segmentation, [20], where atrous convolutions are used in dense prediction tasks to robustly segment objects at multiple scales, and very recently YOLACT++ [21], an instance segmentation algorithm designed to be faster than any previous approaches while maintaining a competitive segmentation performance. While most existing methods perform ship detection on images by locating them using a rectangular bounding box, in recent years some works have performed ship segmentation down to the pixel level, as this approach calculates the ship silhouette, providing some useful shape information that cannot be obtained from detection bounding boxes alone. In [22], a modified version of the U-Net architecture—Squeeze and Excitation U-Net—is presented and trained with a different loss function to increase the intersection over union (IoU) score. Training is performed and the results are obtained using the Airbus ship dataset [6], a large annotated dataset containing remote sensing images of ships with the sea and harbor in the background that was provided on Kaggle for the Airbus ship detection challenge. In [23], a detection and segmentation method for ship detection in satellite remote sensing images is introduced. The authors use a modified version of Mask R-CNN to accurately detect and segment ships at a pixel level. They add a bottom-up path to the FPN structure between the lower layers and the top layer, allowing information from the lower layers to propagate to the top layer. Besides, channel-wise and spatial attention mechanisms are used in the bottom-up path to assign a corresponding weight at each pixel in the feature maps. This allows for a better feature map response to the target features. Like the previous work, the results are obtained by resorting to the Airbus ship dataset. More recently, a two-stage cascade model for detection and segmentation was introduced in [24]. The first stage uses YOLO v3 [25], a detection network, to generate the proposed regions (bounding boxes) from a full image. Applying a detection algorithm before the segmentation stage allows for the removal of undesirable parts of the image such as background areas, improving the overall speed of the model. The second stage resorts to a U-Net network to segment the proposed regions, that typically amounts to only around 5% of the full images: this effectively increases the overall processing speed. Training resorts to the Airbus ship dataset and also to the Seagull dataset, a dataset for Airborne Maritime Surveillance Environments that was created in the context of the development of an intelligent maritime surveillance system using different types of optical sensors [4].

To our knowledge, using temporal correlations for fast ship segmentation in maritime aerial images has never been addressed. In principle, 3D segmentation networks like the V-Net [26], typically used in volumetric medical image segmentation, like magnetic resonance imaging, could be employed for video segmentation, where the third dimension amounts to time. Additionally, some other recent methods, like [27], that explicitly consider the temporal consistency among video frames during training, show promising results regarding temporal coherence between consecutive images in an airborne video sequence. However, these methods do not perform fast enough for real-time inference on embedded hardware on board the UAV and, in the latter case, require a large volume of annotated video sequences for training the network.

Training a segmentation network that takes into account temporal correlations requires temporal sequences of annotated images on a per-pixel basis. There are a few maritime datasets publicly available: in [28] the authors describe the generation of boat classification datasets containing images of boats automatically extracted by the ARGOS system, operating 24/7, in Venice, Italy, to be added to the Maritime Detection, Classification, and Tracking benchmark (MarDCT), a public collection of images and video sequences in maritime scenarios, aiming for evaluating the performance of automatic video surveillance systems for detection, classification and tracking. In [29], a large-scale image dataset for maritime vessels is introduced. The Maritime Vessels (MARVEL) dataset consists of 2 million user-uploaded images of ships and their main attributes like name, flag, length and others. The vessels are sorted into 109 different types and 26 superclasses. Since MARVEL is a dataset for fine-grained visual categorization the only annotations available are category labels, making it suitable for classification problems. The Maritime Satellite Imagery dataset (MASATI) provides maritime scenes of optical aerial images in the visible spectrum obtained from Microsoft Bing maps [2]. The dataset is composed of 6212 satellite images manually labeled according to the following seven classes: land, coast, sea, ship, multi, coast-ship and detail. The labels consist of bounding box annotations for the precise location of vessels. The Singapore Maritime Dataset [30] is created to complement the flaws of MarDCT. It contains images from Singapore waters and, while MarDCT is made of urban navigation scenarios with only one or two boats per image, this dataset is composed of onshore and onboard videos in multiple scenarios. The labels consist of bounding boxes and an identification number, making this a suitable dataset for object detection and tracking algorithms. A new large-scale dataset of ships, SeaShips, is introduced in [31]. It consists of more than 30 thousand images covering six common ship types, with manually annotated bounding boxes and ship categories for every object, taken from real-world video segments acquired by monitoring cameras in a monitoring system along the coastline. In [32], an inshore and offshore maritime vessel detection dataset (ABOship) is introduced that takes into account different factors and scenarios such as background variation, atmospheric conditions, illumination, visible proportion, occlusion and scale variation. It consists of almost 10 thousand images scattered by vessel instances (including nine types of vessels), seamarks and miscellaneous floaters, producing more than 40 thousand annotations containing the class of each object. The Fine-Grained Ship in Remote Sensing dataset (FGSRS-42) [33] comprises 42 categories in more than 9 thousand images. The dataset is formed by a previous work dataset allied to images containing warships and civilian ships at various scales from Google Earth and other popular remote sensing datasets. Image sizes are considerably larger than previous works to suit fine-grained classification. Additionally, images from different spatial resolutions with instances from various aspect ratios contribute to the diversity of the dataset. The annotations contain the name of each image and a class label. The Seagull dataset [4] provides a public collection containing thousands of samples captured from an UAV in challenging situations close to real-world scenarios, with a strong presence of glare, wave crests, wakes, variation of perspective, and objects of interest of different types, scales and shapes, acquired from a diversity of onboard sensors capable of operating in the visible, near-infrared (NIR) and infrared spectrum, providing manually annotated bounding boxes for some of these images. Finally, the Airbus ship dataset [6], a large annotated dataset containing remote sensing images of ships with sea and harbor in the background that was provided on Kaggle for the Airbus ship detection challenge. It provides annotated foreground segmentation masks consisting of rotated rectangles. To our knowledge, however, annotated videos with segmented ships in maritime scenarios, which are required to train segmentation networks that operate on video sequences, are very difficult to find.

In this paper, a two-stage approach for ship segmentation from airborne images that takes into account temporal correlations between frames is proposed. In the first stage, the YOLACT++ network performs very fast ship detection and segmentation on each frame, independently of each other frame. Since segmentation results provided by this stage can be inconsistent between consecutive images, in the second stage a post-processing technique based on Conditional Random Fields (CRFs) [34] ensures temporal correlation between the current frame and the previous ones. The overall architecture is presented in Figure 3.

Additionally, since annotated maritime video datasets for object segmentation are very scarce and difficult to obtain, we also present a novel Maritime Synthetic (MarSyn) dataset to train and test the system in video sequences with automatic generation of ground truth annotations, allowing a significant cost reduction related to real-world data acquisition and hand-made image annotation. The contributions of this paper are, therefore,

A two-stage approach for real-time ship segmentation from airborne images that improves the temporal coherence between consecutive video frames;A novel Maritime Synthetic dataset that can be used to train and test segmentation methods on maritime scenarios, that can be easily extended while providing automatic ground truth annotations.

The remainder of this document is organized in the following manner: In Section 2, we present the developed maritime synthetic dataset, the segmentation network used, and the post-processing stage based on 3D Conditional Random Fields. In Section 3, we describe the experimental results obtained using the proposed approach, and in Section 4, we provide some concluding remarks.

## 2. Materials and Methods

In order to train the proposed learning architecture with sufficiently rich sequential data, corresponding to airborne maritime surveillance RGB images, first the novel synthetic dataset developed in this work is described in Section 2.1. After that, the chosen segmentation network, Yolact++ [21], is briefly described in Section 2.2. In Section 2.3, the post-processing stage that resorts to 3D fully connected CRFs to ensure temporal coherency is described in detail, and finally, in Section 2.4, we present the datasets used in this work.

### 2.1. Maritime Synthetic Dataset

Training a segmentation network for maritime surveillance video sequences requires large amounts of labeled data with foreground masks corresponding to ships. As discussed before, this type of dataset is very difficult to obtain, and thus a synthetic maritime dataset will be developed in this work for this purpose.

Synthetic data is a method of data generation for machine learning that creates an entirely artificial dataset. It can use real-world data, retaining all the insight, or it can be created from scratch, allowing a synthetic-to-real adaptation. Therefore, it is possible to train a model on synthetic data and transfer the results to real-world data, a concept that is correlated with data augmentation [35]. Additionally, if automatically created it leaves manual labeling out of the equation, reducing human errors and increasing the quality of the annotations.

Creating this data from scratch requires 3D modeling software. In [36], the authors found it almost impossible to acquire imagery from spacecrafts photographed by other spacecrafts in outer space. To generate the data, they resorted to Blender [37], an open-source software for 3D modeling that can generate large numbers of photo-realistic images that include meta-information as image labels. The dataset is composed of 60 thousand images of four different unmanned NASA spacecrafts. In [38], the authors use synthetically rendered images of drinking glasses to train deep neural networks on object detection. Additionally, by having a real-world dataset of drinking glasses, they are able to train the same network with it and compare results. The experiments have shown that the use of synthetic data resulted in comparable results, even though real-world data outperforms synthetic data.

The Maritime Synthetic (MarSyn) dataset developed in this work consists of 25 different photo-realistic video sequences, each one containing 1000 frames, obtained using Blender. Its 25 thousand images and their corresponding 34 thousand annotations aim to simulate multiple maritime scenarios and conditions such as sunny, cloudy and sunset environments, near coast images and reflections on the water. Different types of vessels are deployed in the simulations, which increases the diversity of the dataset, publicly available upon request (https://vislab.isr.ist.utl.pt/marsyn-dataset/, accessed on 23 September 2022).

Figure 4 shows the different vessels present on the MarSyn dataset. The vessels vary in type (cargo ships, military ships, fishing boats, speed boats, rescue rafts and others), length (from 3 m all the way up to 125 m), shape and color. The simulated camera is placed and moved accordingly to an UAV and it captures imagery at heights between 150 and 1000 m. All of the captured images are RGBA and are available in two different resolutions, 1280 × 720 pixels and 550 × 550 pixels. An example of an image and its corresponding annotations are shown in Figure 5. This method of maritime images generation allows the creation of rich and diverse maritime datasets, with different scenarios and configurations, removing the costs of real-world data acquisition and making it possible to generate specific situations that can be hard to capture in real-world scenarios: in this way, it is possible to train a network to segment different types of vessels without having to capture them in the real world. Additionally, by producing the foreground masks automatically, no manual labeling is required, reducing time and human errors. Some footage presenting the different MarSyn dataset scenarios is available for illustration purposes (https://youtu.be/gHr8gepvG3g, accessed on 23 September 2022).

### 2.2. Segmentation Network

The first stage of the proposed method is used to detect and segment vessels on each image in an independent way. We use Yolact++ [21], a recent instance segmentation algorithm designed to be faster than any previous state-of-the-art approaches and that is already being used for various kinds of segmentation tasks [39,40]. Its speed is achieved by breaking down instance segmentation into two parallel subtasks: (1) generating a set of prototype masks and (2) predicting per-instance mask coefficients. The backbone detector is designed to prioritize feature richness and speed by closely following RetinaNet [41] architecture. It is composed, by default, of a combination between a ResNet-101 network [42] and an FPN [43]. The base image size is 550 × 550 pixels allowing for consistent evaluation times per image. Figure 6 shows Yolact++ architecture.

To train this model, three different losses are combined: classification loss, box regression loss and mask loss, with a weight of 1.0, 1.5 and 6.125, respectively. The first two losses are defined in the same way as in the Single shot multibox detector [44], while the mask loss is computed through the pixel-wise binary cross-entropy between the predicted mask and the ground truth mask.

In [21], the authors experimented multiple backbones configurations both on the Yolact architecture and the latest version Yolact++. Table 1 displays these results for different architectures on the MS COCO dataset, where it is visible that every configuration achieves real-time segmentation when inference is performed on an Nvidia TITAN Xp GPU.

### 2.3. Post-Processing Using 3D Fully Connected CRFs

Conditional Random Fields (CRFs) is a class of discriminative models that takes the contextual information of neighbor variables to obtain a prediction for the current variable, by modeling the dependencies between variables as a graphical model. In image processing, this typically amounts to increase the influence of nearby pixels on the final prediction for a given pixel: this implicitly defines a spacial smoothness prior over the image. To achieve this, usually a unary potential and a pairwise potential are defined. The unary potential is computed on individual pixels and denotes the potential of a given pixel to belong to a certain class, while the pairwise potential is computed between a pair of pixels and denotes the potential of those pixels to belong to the same class [34]. After defining the potential function, an optimization procedure tries to maximize this function over the class probabilities for each of the pixels. There are two major classes of CRFs, sparse and dense: while sparse CRFs consider only connections between nearby pixels, dense CRFs present global connectivity. As a consequence, dense CRFs are much harder to optimize, even if they provide, in principle, better segmentation results.

In this paper, we propose a 3D fully connected CRF capable of making predictions for one image frame considering information from previous image frames in the same video sequence. We adapt the 3D fully connected CRF [45] designed for tasks of spacial brain lesion segmentation in multi-channel MRIs, by modifying this model to receive a volume of RGB images corresponding to a video sequence instead of a volume of grayscale images corresponding to a full MRI scan, and we fix the number of channels to 3, to accommodate the corresponding RGB channels. The proposed 3D fully connected CRF model minimizes an energy function given by
(1)$$E\left(z\right)=\sum_{i}\psi_{u}\left(z_{i}\right)+\sum_{ij,i\neq j}\psi_{p}\left(z_{i},z_{j}\right)\;,$$
where *z* represents the labels of every pixel on the image. The first term is computed independently for each pixel by a classifier and expresses the unary potential, according to
(2)$$\psi_{u}\left(z_{i}\right)=-\log P(z_{i})\;.$$

The unary potential is given by the negative log-likelihood, where $P(z_{i})$ is the label probability at pixel *i* given by the segmentation network. The second term in Equation (Equation 1) expresses the pairwise potential by the following relation:(3)$$\psi_{p}\left(z_{i},z_{j}\right)=\mu \left(z_{i},z_{j}\right)\;\cdot \;\sum_{m=1}^{M}w^{m}\;\cdot \;k^{m}\left(f_{i},f_{j}\right)\;,$$
where μ is a simple label compatibility function given by the Pott’s Model, $\mu \left(z_{i},z_{j}\right)=\left[z_{i}\neq z_{j}\right]$. It introduces a penalty for nearby pixels assigned with different labels. Each $k^{m}\left(f_{i},f_{j}\right)$ represents a Gaussian kernel between pixel features given by
(4)$$w_{1}\exp\left(-\sum_{d=x,y,z}\frac{|p_{i,d}-p_{j,d}{|}^{2}}{2\sigma_{\alpha ,d}^{2}}\right)+w_{2}\exp\left(-\sum_{d=x,y,z}\frac{|p_{i,d}-p_{j,d}{|}^{2}}{2\sigma_{\beta ,d}^{2}}-\sum_{c=r,g,b}\frac{|I_{i,c}-I_{j,c}{|}^{2}}{2\sigma_{\gamma ,c}^{2}}\right)\;,$$
where for multi-class image segmentation and labeling, two contrast-sensitive potential kernels are used, with weights $w_{1}$ and $w_{2}$. These kernels operate in the feature space defined by the pixel position coordinates $p_{i,d}$ and the different intensities of the pixel RGB channel $I_{i,c}$. The first term in Equation (Equation 4) is the smoothness kernel, and it is used to remove small isolated regions, i.e., it discourages isolated tiny segmented regions of noisy pixels. The parameter $\sigma_{\alpha ,d}$ controls the degree of pixel nearness on the three spatial coordinates (*x*, *y* and *z*). The second term of the equation is the appearance kernel, being inspired by the observation that nearby pixels with similar colors are likely to be in the same class. The degrees of nearness and similarity are controlled by the parameters $\sigma_{\beta ,d}$ and $\sigma_{\gamma ,c}$ over the three spatial coordinates (*x*, *y* and *z*) and the three RGB channels (*r*, *g* and *b*), respectively.

### 2.4. Datasets

Besides the synthetic MarSyn dataset, we also use the Seagull and Airbus datasets for training and testing: some sample images from these datasets can be seen in Figure 1 and Figure 2, respectively. Samples from the Seagull dataset are manually labeled for segmentation purposes, while those belonging to the Airbus dataset have rectangular annotations already provided. T1 is the same training set used in [24] and is introduced for comparison purposes with that work. T2 is a training set with synthetic images only. For evaluation, we built three test sets, S1, S2 and S3, with increasingly demanding scenarios for segmentation: while S1 comprises images with only one object of interest, typically with a medium/large size in the image, S3, on the other hand, images with multiple objects of interest with medium/small size. S2 corresponds to an intermediate situation in terms of the complexity of the image to be segmented. Images in S4 are taken from [24] and once again are used only for comparison purposes. Finally, S5 and S6 correspond to temporal correlated images taken from video sequences, one from the Seagull dataset and the other from the MarSyn dataset. Table 2 summarizes the datasets used in this work.

## 3. Results

In this section, an experimental evaluation of the proposed approach is performed: we evaluate different backbones for the Yolact++ network, then we compare this network results with the ones obtained in [24], showing the performance increase when the post-processing 3D CRF is used and, finally, we present some qualitative results regarding the usefulness of this method when single frames are lost or when some outlier results are obtained during the segmentation phase. For evaluating the segmentation performance, we use the standard Intersection over Union (IoU) score, also known as the Jaccard similarity coefficient, by considering the pixelwise intersection and union of the predicted and ground truth ship masks on the images. All tests are run on a PC with a Intel Core i7-9700F @3.00 GHz, 32 GB RAM, equipped with an Nvidia GeForce GTX 2080 GPU with 8 GB VRAM.

### 3.1. Yolact++ Backbone

To find the optimal Yolact++ configuration, we started by testing the two different backbone architectures available, ResNet50 and ResNet101. We trained this network with both backbones using training set T1 and we then tested both architectures on the three available Airbus test sets, S1, S2 and S3, corresponding to increasingly difficult to segment images. We also evaluated the performance on a global test set comprising the previous test sets. Table 3 shows the segmentation mask IoU results and the maximum frame rate achieved by these two backbones.

As expected, the lighter ResNet50 architecture achieves a better segmentation rate of 29 frames per second (FPS) when compared to the ResNet101 backbone. Since segmentation performance, measured by the IoU score, is similar for both backbones, we chose the ResNet50 backbone in the following experiments. A typical learning curve for this model is presented in Figure 7, where the training loss and the validation set IOU are presented as a function of the number of training epochs in the T1 and T2 datasets. We also present the corresponding precision–recall curve in Figure 8.

### 3.2. Segmentation Results

After training the Yolact++ network using training set T1, we compared its results to the results obtained in a previous related work [24], obtaining the IoU and FPS scores presented in Table 4. Both networks are trained and tested on the same datasets (T1 and S4, respectively).

While the cascade model introduced in [24] outperforms the Yolact++ network in terms of segmentation mask IoU, the Yolact++ is almost three times faster than the cascade model in terms of inference speed. However, we believe that the results obtained in [24] may result from a slightly overfit to the training data, as both the training and test sets contain mostly satellite images coming from the Airbus Detection Dataset. This dataset assigns ground truth rotated rectangular masks to its images, which do not exactly match the true ship contour, as depicted in Figure 9.

We believe the U-Net used in [24] is overfitting these rectangular masks, as it is trained from scratch with this data. Yolact++, on the other hand, since it uses this data to refine a model pre-trained on the COCO dataset, is not as sensitive as the cascade model to this kind of annotation, and thus achieves a lower score when evaluated on a test set comprising these rectangular ground truth masks.

### 3.3. Post-Processing Using 3D CRFs

The 3D Conditional Random Field used in this work as a post-processing stage has a number of hyper-parameters that need to be fixed, namely $w_{1}$, $w_{2}$, $\sigma_{\alpha }$, $\sigma_{\beta }$ and $\sigma_{\gamma }$. To find the values first we trained the Yolact++ network with MarSyn dataset T2, and then ran this network over the test set S6, consisting of 25 synthetic video sequences with 30 images each. We used the resulting segmented images and the corresponding ground truths to obtain the parameters values that maximized Equation (Equation 4). First, we fixed the temporal window for the 3D CRF to the current frame and the previous four frames, due to computational constraints and inference speed, and then we optimized the values corresponding to the 3D fully connected CRF by performing an exhaustive grid search over these parameters, considering all combinations of integer values from 1 to 10 assigned to each of these 5 parameters. In the end, we obtained $w_{1}=1$, $w_{2}=5$, $\sigma_{\alpha }=1$, $\sigma_{\beta }=4$ and $\sigma_{\gamma }=5$.

We compared the performance of the segmentation system with and without the post-processing stage using 3D CRFs, on the synthetic MarSyn test set S6 and on the Seagull test set S5. The results are summarized on Table 5. We can see that the segmentation performance increases; the processing framerate, however, significantly drops.

Figure 10 and Figure 11 present some qualitative results obtained on MarSyn detection and Seagull dataset, respectively.

We can notice that the use of the post-processing stage based on 3D CRF visually improves the ship contour and increases its detail and accuracy, making it easier to distinguish between different type of vessels using the final segmentation mask alone.

To show the usefulness of using the 3D CRF as a post-processing stage we simulated a situation where some image frames are lost, as depicted in Figure 12.

The first column corresponds to four images taken from a video sequence that captures a ship in a maritime scenario, while the second column presents the corresponding ground truth annotations. In the third column, the Yolact++ network is used to segment these images, and we simulate a case where the segmentation network misses to predict the ship pixels on the second frame, resulting in a prediction without any ship pixels. Adding a 3D fully connected CRF after the segmentation network allows us to not only improve the segmentation results, but also to add temporal continuity to the system. The temporal continuity is introduced by taking into account information from the previous four frames and predictions on the CRF prediction process. As shown in the forth column, despite this missing frame, the CRF is able to reconstruct the object based on previous frames, thus acting as a temporal filter, and the final segmentation result provides the ship contour for all frames.

This kind of filtering is also useful in the presence of outliers or incorrect segmentations performed in the previous stage. In Figure 13, we deliberately manually changed the Yolact++ segmentation results, by over-segmenting the first and fifth frames, by adding a second ship on the second frame, and by providing a very small segmentation outlier on the third frame. Despite these disturbances, the final result is very stable due to the temporal filtering introduced by the 3D CRF, as illustrated in the demonstrating video (https://youtu.be/ujKuQDYUps0, accessed on 23 September 2022).

## 4. Discussion

In this paper, we introduced a new maritime surveillance synthetic dataset that can be used for training of ship segmentation networks: to our knowledge, this is the first publicly available artificial dataset consisting of synthetic images simulating maritime surveillance scenarios. It provides high quality ground truth annotations automatically extracted during image generation, which eliminate human labeling errors. It can also be easily extended to provide a much larger volume of training data and to accommodate other types of ships, different atmospheric conditions and unusual camera perspectives.

The second contribution is a novel segmentation method that takes into account image temporal correlations, by passing the segmentations obtained in a first stage through a 3D CRF that acts as a post-processing stage. These two stages are completely independent from each other: we used the state-of-the-art Yolact++, that provides quality segmentation with real-time inference capabilities, but in principle any other segmentation network can be used. Our experimental results show very promising results and the ability of this post-processing stage to cope with missing frames, segmentation errors occurring in the segmentation stage and other temporal inconsistencies between consecutive frames. One drawback of this method is its computational burden, making the framerate significantly drop when compared to the use of a segmentation network alone, as can be seen on Table 5, even if the final framerate is still manageable. One of the reasons for this to happen is the fact that currently the 3D CRF runs on the CPU, while segmentation networks usually use the massive parallelism capabilities of the GPUs where the inference is done. As future work, we intend to optimize this second stage to significantly reduce its inference time. One approach we intend to follow is to analyze how a sparse CRF performs in this situation: currently, a dense 3D CRF is used, where connections between every pair of pixels on the current and previous images are considered. This is very CPU intensive, and if we manage to reduce the set of connections available, we can expect a huge performance increase regarding the post-processing speed. We also intend to study the influence of synthetic data in the overall segmentation performance. It is not trivial to get the best ratio of synthetic to real-world data, and the results can significantly change depending on the test set used to evaluate the final segmentation performance. This deserves a thoroughly investigation. 

## Figures and Tables

**Figure 1 sensors-22-08090-f001:**
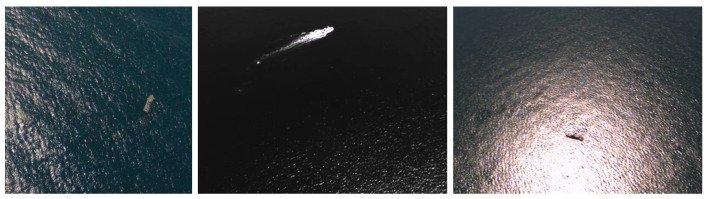
Example of airborne maritime surveillance images taken from the Seagull dataset [4].

**Figure 2 sensors-22-08090-f002:**
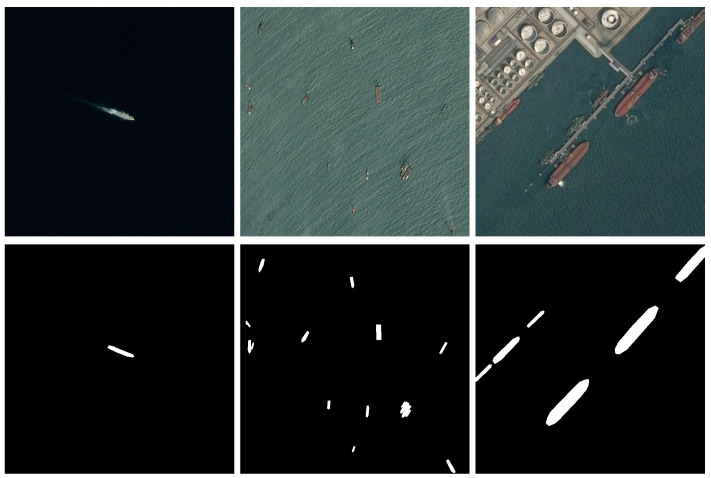
Example of segmentation of maritime surveillance images, taken from the Airbus ship detection dataset [6]. Top: original images. Bottom: corresponding ship segmentation masks.

**Figure 3 sensors-22-08090-f003:**
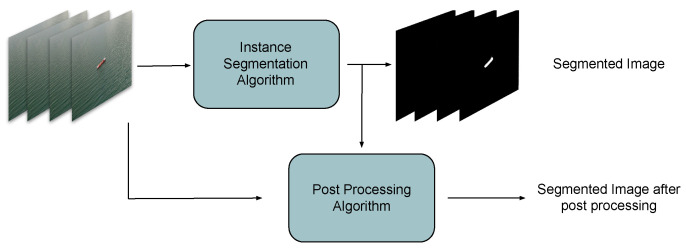
Overall system architecture.

**Figure 4 sensors-22-08090-f004:**
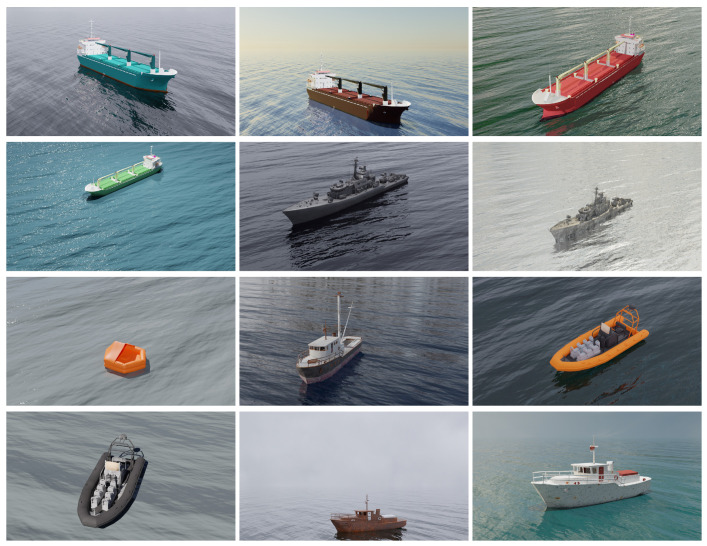
Different types of vessels present on the MarSyn dataset.

**Figure 5 sensors-22-08090-f005:**
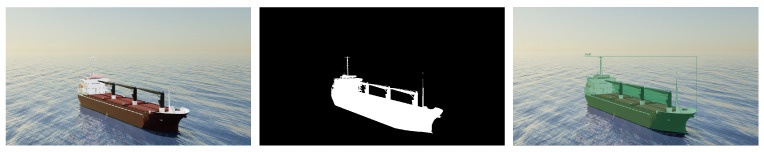
Example of MarSyn dataset annotations. Left: original image; Center: foreground mask; Right: COCO format annotation.

**Figure 6 sensors-22-08090-f006:**
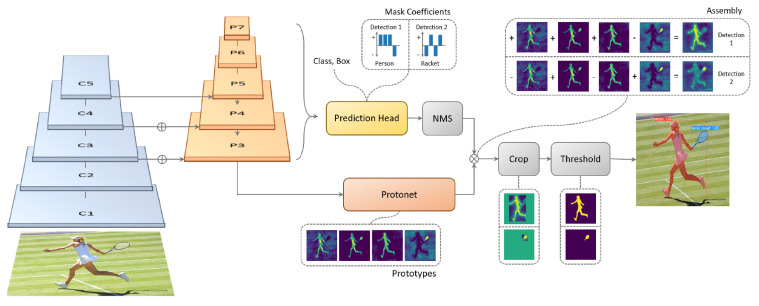
Yolact++ Architecture (Image reproduced from [21] with permission from the authors).

**Figure 7 sensors-22-08090-f007:**
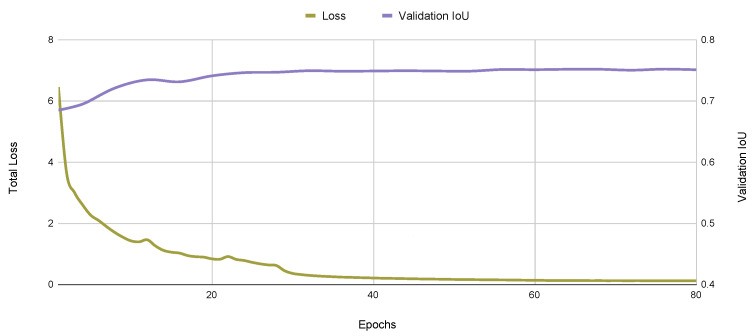
Typical learning curve for the Yolact++ algorithm, showing the evolution of training loss and the validation IOU.

**Figure 8 sensors-22-08090-f008:**
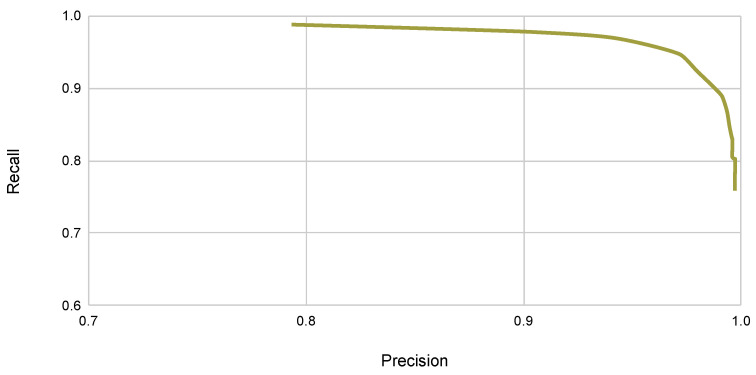
Typical precision–recall curve for the Yolact++ algorithm on the maritime dataset.

**Figure 9 sensors-22-08090-f009:**
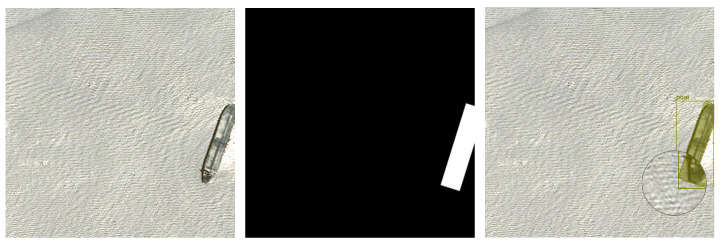
Airbus Ship Detection dataset ground truth annotation example. Left: original image; Middle: ground truth annotation mask. Right: magnified detail showing the discrepancy between the mask and the true ship segmentation.

**Figure 10 sensors-22-08090-f010:**
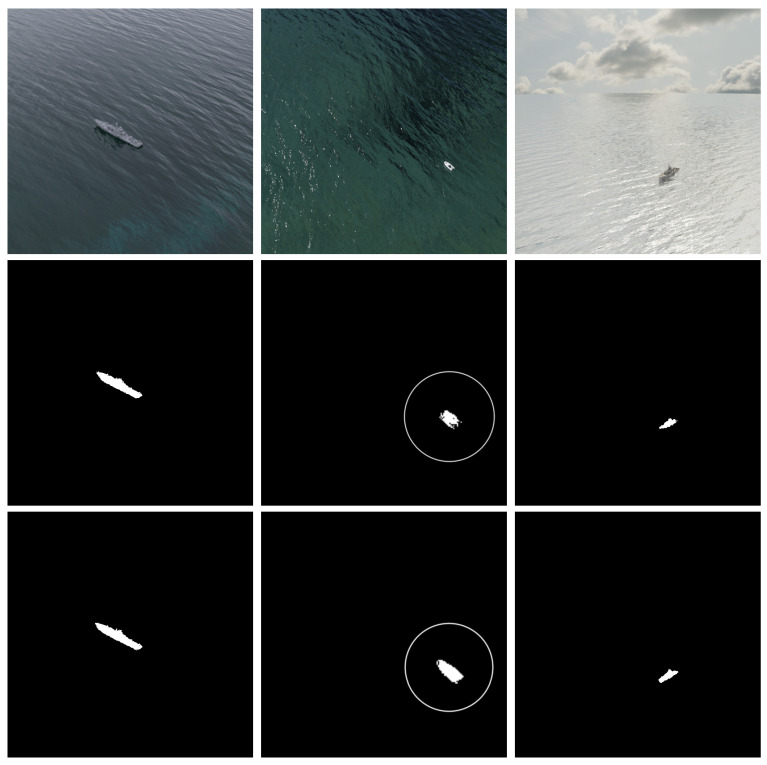
Qualitative evaluation on the MarSyn detection dataset. Top row: original images; middle row: Yolact++ segmented images; bottom row: final segmentation result after post-processing with 3D CRF. Circles in the second column denote a magnified image for better visualization.

**Figure 11 sensors-22-08090-f011:**
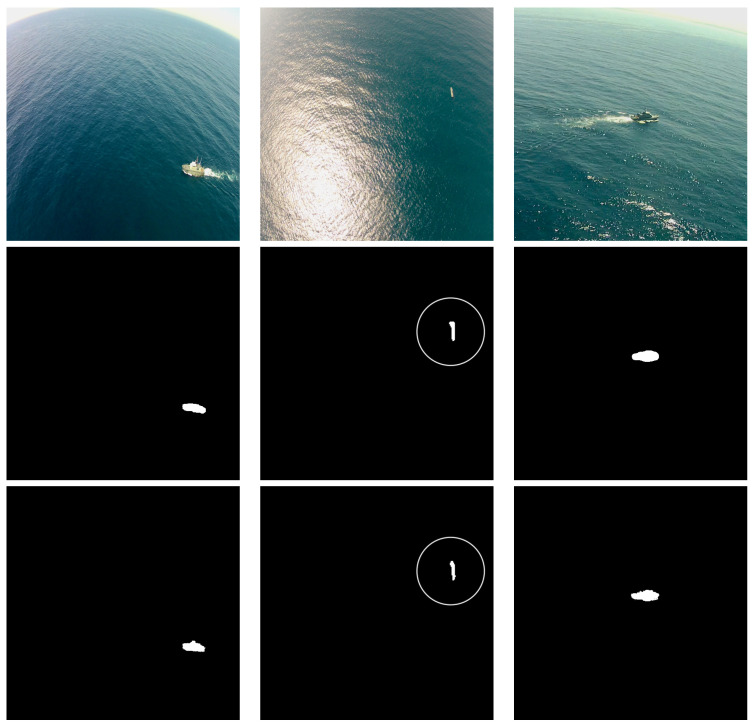
Qualitative evaluation on the Seagull dataset. Top row: original images; middle row: Yolact++ segmented images; bottom row: final segmentation result after post-processing with 3D CRF. Circles in the second column denote a magnified image for better visualization.

**Figure 12 sensors-22-08090-f012:**
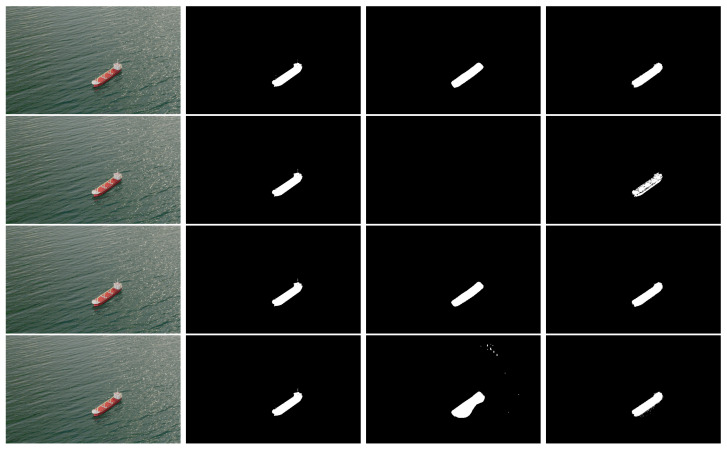
Simulated frame loss on the MarSyn dataset. First column: original images taken from a video sequence, from older (top) to newer (bottom); second column: corresponding ground truth masks; third column: Yolact++ segmented images, where the loss of the second frames of the sequence is simulated; fourth column: final segmentation result after post-processing with the 3D CRF.

**Figure 13 sensors-22-08090-f013:**
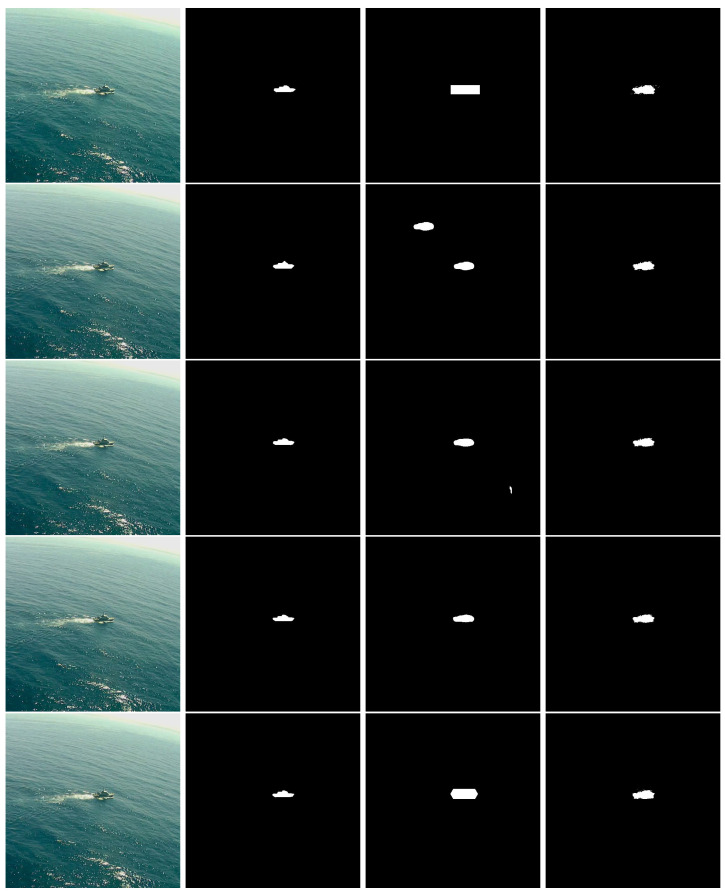
Simulating incorrect Yolact++ segmentations on the Seagull dataset. First rows correspond to older images, while the bottom row corresponds to the most recent image frame. First column: original images; second column: ground-truth annotations; third column: Yolact++ manually disturbed segmented images; fourth column: final segmentation result after post-processing with the 3D CRF.

**Table 1 sensors-22-08090-t001:** Results of different backbone architectures on MS COCO [21].

Network	Image Size	Backbone	FPS	mAP (%)
Yolact	550	ResNet50-FPN	42.5	28.2
Yolact	550	DarkNet53-FPN	40.0	28.7
Yolact	550	ResNet101-FPN	33.5	29.2
Yolact	700	ResNet101-FPN	23.6	31.2
Yolact++	550	ResNet50-FPN	33.5	34.1
Yolact++	550	ResNet101-FPN	27.3	34.6

**Table 2 sensors-22-08090-t002:** Different datasets used in this work.

Name	Usage	Source	# Images	Type of Labels	Video Sequence?
T1	Training	Airbus dataset	19,800	Provided	No
		Seagull dataset	200	Manual	
T2	Training	MarSyn dataset	10,000	Automatic	No
S1	Testing	Airbus dataset	200	Provided	No
S2	Testing	Airbus dataset	200	Provided	No
S3	Testing	Airbus dataset	200	Provided	No
S4	Testing	Airbus dataset	100	Provided	No
S5	Testing	Seagull dataset	750	Manual	Yes
S6	Testing	MarSyn dataset	750	Automatic	Yes

**Table 3 sensors-22-08090-t003:** Segmentation mask IoU for the different backbone architectures.

Backbone	IoU				FPS
	S1	S2	S3	Global	
ResNet50	0.893	0.831	0.789	0.828	29
ResNet101	0.892	0.830	0.790	0.828	23

**Table 4 sensors-22-08090-t004:** IoU score and inference speed.

Network	IoU Score	FPS
Cascade model [24]	0.94	11
Yolact++	0.79	29

**Table 5 sensors-22-08090-t005:** Segmentation results with and without 3D CRF post-processing stage.

Architecture	IoU		FPS
	S5 (Seagull)	S6 (MarSyn)	
Yolact++	0.771	0.796	29
Yolact++/3D CRF	0.786	0.833	5

## Data Availability

Access to the MarSyn dataset can be requested at https://vislab.isr.ist.utl.pt/marsyn-dataset/ (accessed on 23 September 2022). Implementation details regarding the proposed architecture can be accessed at https://github.com/MiguelAngeloMartinsRibeiro/Real-Time-Ship-Instance-Segmentation-with-3D-fully-connected-CRF-for-Maritime-Surveillance (accessed on 23 September 2022).

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
