# Peer review of "Real-Time Ship Segmentation in Maritime Surveillance Videos Using Automatically Annotated Synthetic Datasets"

_sensors, 2022, doi:10.3390/s22218090_

Round 1

Reviewer 1 Report

in this manuscript, the authors proposed a synthetic dataset boosted real-time segmentation for maritime surveillance videos. The hyperthesis and result seem to be consistent. According to the results, the proposed methods achieve a high IoU compared to methods without 3D synthesis and post-processing. Before it is published, I have the following question.

(1)As depict in line 280-282, the authors use the state of the art method Yolact++, but as shows in paper with code website, the state of the art of instance segementation is not Yolact++,Could you try the real state of the art instance segmentation model in your paper?

(2) According to Table 2. could you please add some discussion about the effect of the different ratio of synthetic dataset and manually labeled dataset, in addition please add the training error verses the epoch in your paper to demonstrate the converge speed of your model.

(3) As descript in 386 to 387, could you demonstrate all the grid search result, and the scope of your search space. 

Reviewer 2 Report

Abstract needs to improve with technical language.

Authors have written introduction and related work/literature survey in one section. It should be different which can improve the readability of the manuscript.

Figure 6 resolution needs to improve and it is taken from the reference no. 20.

Authors must emphasize on novelty of the work and discuss proposed workflow or architecture.

In section 3, author should write the software/ libraries used for this work.

Authors have compared and discuss the results of different stages of Yolact++ algorithm. However, comparison of result analysis is required with the existing algorithm. Authors have compared with only one algorithm. Detail result analysis is expected. This can show the efficiency of the proposed work.

Result analysis with accuracy, precision-recall curve of the proposed work has to be presented.

Conclusion and future work is missing. They must be added.

Round 2

Reviewer 2 Report

The comment-2 is for the increasing the readability of the manuscript. It is general way to define sections in manuscript.

I do agree with the justification of comment – 4 by author and the given justification is already mentioned in manuscript. However, there must be a workflow diagram or algorithm of proposed work which will be easy to understand the proposed work and improve the readability. (Just like YOLACT++ - figure 6, such figure or algorithm of proposed work should present in the manuscript.)

Conclusion and future work should be explicitly written. It should not be included in discussion. This is also a part of section in any manuscript. This is general guideline. It should be done to improve readability.
